# Genetic Variants Associated with Longitudinal Cognitive Performance in Older Breast Cancer Patients and Controls [note 1]

**DOI:** 10.3390/cancers15112877

**Published:** 2023-05-23

**Authors:** Kelly Nudelman, Kwangsik Nho, Michael Zhang, Brenna C. McDonald, Wanting Zhai, Brent J. Small, Claire E. Wegel, Paul B. Jacobsen, Heather S. L. Jim, Sunita K. Patel, Deena M. A. Graham, Tim A. Ahles, James C. Root, Tatiana Foroud, Elizabeth C. Breen, Judith E. Carroll, Jeanne S. Mandelblatt, Andrew J. Saykin

**Affiliations:** 1Department of Medical and Molecular Genetics, Indiana University School of Medicine, Indianapolis, IN 46202, USA; 2Indiana Alzheimer’s Disease Research Center, Indiana University School of Medicine, Indianapolis, IN 46202, USA; 3Indiana University Genetics Biobank, Indiana University School of Medicine, Indianapolis, IN 46202, USA; 4Center for Neuroimaging, Department of Radiology and Imaging Sciences, Indiana University School of Medicine, Indianapolis, IN 46202, USA; 5Melvin and Bren Simon Comprehensive Cancer Center, Indiana University School of Medicine, Indianapolis, IN 46202, USA; 6Lombardi Comprehensive Cancer Center, Georgetown University, Washington, DC 20057, USA; 7H. Lee Moffitt Cancer Center and Research Institute, Tampa, FL 33612, USA; 8School of Aging Studies, University of South Florida, Tampa, FL 33620, USA; 9Division of Cancer Control and Population Studies, National Cancer Institute, Bethesda, MD 20892, USA; 10Department of Population Sciences, City of Hope Comprehensive Cancer Center, Duarte, CA 91010, USA; 11John Theurer Cancer Center, Hackensack University Medical Center, Hackensack, NJ 07601, USA; 12Department of Psychiatry and Behavioral Sciences, Memorial Sloan Kettering Cancer Center, New York, NY 10065, USA; 13Cousins Center for Psychoneuroimmunology, University of California, Los Angeles, CA 90095, USA; 14Department of Psychiatry & Biobehavioral Sciences, Semel Institute for Neuroscience and Human Behavior, University of California, Los Angeles, CA 90095, USA

**Keywords:** cancer, cognition, GWAS, genetics, CRCD

## Abstract

**Simple Summary:**

While numerous publications have shown that some patients experience cancer- and treatment-related cognitive decline, the role of genetics in risk for cognitive decline has not yet been established. In this study, our goal was to identify genetic factors affecting risk for cognitive decline in older female breast cancer survivors. The study identified several genetic variants and genes that were associated with differences in patterns of cognitive decline in cancer patients compared to controls, suggesting that genetics can play an important role in modifying risk for cognitive decline in older cancer survivors. It will be important for additional research to replicate these findings in other cancer populations; if validated, these findings could inform therapeutic research, as well as inform evaluations of risk for cognitive decline in older cancer survivors.

**Abstract:**

**Background:** There have been no published genome-wide studies of the genetics of cancer- and treatment-related cognitive decline (CRCD); the purpose of this study is to identify genetic variants associated with CRCD in older female breast cancer survivors. **Methods:** Analyses included white non-Hispanic women with non-metastatic breast cancer aged 60+ (*N* = 325) and age-, racial/ethnic group-, and education-matched controls (*N* = 340) with pre-systemic treatment and one-year follow-up cognitive assessment. CRCD was evaluated using longitudinal domain scores on cognitive tests of attention, processing speed, and executive function (APE), and learning and memory (LM). Linear regression models of one-year cognition included an interaction term for SNP or gene SNP enrichment*cancer case/control status, controlling for demographic variables and baseline cognition. **Results:** Cancer patients carrying minor alleles for two SNPs, rs76859653 (chromosome 1) in the hemicentin 1 (*HMCN1)* gene (*p* = 1.624 × 10^−8^), and rs78786199 (chromosome 2, *p* = 1.925 × 10^−8^) in an intergenic region had lower one-year APE scores than non-carriers and controls. Gene-level analyses showed the POC5 centriolar protein gene was enriched for SNPs associated with differences in longitudinal LM performance between patients and controls. **Conclusions:** The SNPs associated with cognition in survivors, but not controls, were members of the cyclic nucleotide phosphodiesterase family, that play important roles in cell signaling, cancer risk, and neurodegeneration. These findings provide preliminary evidence that novel genetic loci may contribute to susceptibility to CRCD.

## 1. Introduction

Improved early detection and treatments for breast cancer have greatly increased the number of survivors [1,2]. However, cancer and treatment-related cognitive decline (CRCD) has become an increasing concern [3], particularly for individuals over age 65, who are projected to constitute 73% of the anticipated 26.1 million cancer survivors in the United States by 2040 [1]. As age is a risk factor for cognitive decline and dementia, this portion of the cancer survivor population might be more vulnerable to CRCD [1,3,4].

Several studies have linked single nucleotide polymorphisms (SNPs) in candidate genes to CRCD. For example, studies have shown that the *APOE* e4 allele, the major risk factor for Alzheimer’s disease (AD), is associated with worse neurocognitive outcomes in some cancer patients [3,5,6,7]. More recently, there has been some evidence that *APOE* e2 may protect against CRCD in cancer survivors [8]. However, there have not been any published studies using a genome-wide analysis approach to identify loci associated with CRCD, and no genetic studies have focused on older survivors. Genome-wide investigation of CRCD genetic etiology could inform counseling and treatment of patients, as well as research on drugs targeting prevention and treatment of CRCD.

The primary objective of this study was to identify genetic variants showing different associations with longitudinal changes in neuropsychological domain scores for attention, processing speed, and executive function (APE) or learning and memory (LM) in older breast cancer cases and non-cancer controls. These domains were chosen based on previous research into the cognitive effects of cancer and treatment, as changes in these domains have been associated with CRCD [3]. The secondary objective was to identify genes enriched for variants showing differential associations with the APE and LM domains in cases and controls. These analyses aimed to identify variants and genes interacting with breast cancer diagnosis to influence risk of CRCD. This is the largest study of CRCD genetics published to date, and highlights the utility of this approach towards advancing the state of scientific knowledge in this field.

## 2. Materials and Methods

This study was a secondary analysis of specimens and data from the Thinking and Living with Cancer (TLC) study. TLC recruited participants from 13 oncology practices at or affiliated with six national sites: Georgetown University, Memorial Sloan Kettering Cancer Center, Moffitt Cancer Center, City of Hope Comprehensive Cancer Center, Hackensack University Medical Center, and Indiana University School of Medicine. All Institutional Review Boards approved the protocol (NCT03451383).

### 2.1. Study Population

TLC patients were female breast cancer patients (stage 0–3) diagnosed at age 60+ years. Friends of patients were recruited as controls when possible; if not possible, controls were recruited at each site. All controls were frequency matched by age, race/ethnicity, and education by site. Participants are followed prospectively with baseline pre-treatment/enrollment and then annual visits. For this study, baseline and one-year follow-up data were utilized for participants enrolled from 2010 to 2019. The TLC study is ongoing and has been extensively described in other publications [3,7]. Briefly, participants were excluded for a history of stroke, head injury, major Axis I psychiatric disorders, neurodegenerative disorders, ever previously receiving chemotherapy or hormonal therapy, having had active treatment for cancer within the last five years prior to enrollment, or having a Mini-Mental State Examination score < 24 or Wide Range Achievement Test-Fourth Edition (WRAT4) Word Reading score less than third grade level [9,10]. Additional eligibility for this analysis included having a biospecimen for GWAS testing and one-year cognitive data. To avoid bias from genetic ancestry and given the small number of minority participants in the study, genetic analyses were limited to white, non-Hispanic participants.

Of the 807 participants with processed genetic data passing quality control, 142 (95 cases, 47 controls) were missing clinical, demographic, or cognitive data, and were not included in the final analyses. These excluded participants were similar in age (mean = 68.13, standard deviation = 6.6), education (mean = 15.1, standard deviation = 2.3) and WRAT4 score (mean = 109.8, standard deviation = 14.2) compared to participants included in the analysis. A total of 665 individuals including 325 cases and 340 controls with genetic data and cognitive performance domain data were included in the analyses (See Figure 1 CONSORT Diagram).

### 2.2. Data Collection

The baseline visit for TLC participants included collection of blood or saliva. In cases where a sample could not be collected at baseline, samples were collected at follow-up visits. Baseline assessments in patients were conducted following cancer-related surgery, but prior to initiation of chemotherapy, radiation, or hormone treatments.

Thirteen neuropsychological tests were administered at each visit to obtain data for two per protocol pre-specified cognitive domains: attention, processing speed, and executive function (APE), and learning and memory (LM). The APE domain score includes the Digits Forward and Backward subtests from the Neuropsychological Assessment Battery (NAB), Trail Making Tests A and B, the Controlled Oral Word Association Test, and the Digit Symbol subtest from the Wechsler Adult Intelligence Scale-III [11,12,13,14,15]. The LM domain includes the Logical Memory I and II subtests from the Weschler Memory Scale-III and the Immediate Recall, Short Delayed Recall, and the Long Delayed Recall scores from the NAB List Learning Test [13,14,15]. Raw scores were standardized using the control group mean and standard deviation at baseline stratified by age and education. Standardized scores were then used to calculate z-scores for each domain for every participant, as described previously [8,16]. WRAT4 reading scores were obtained at baseline [9].

Collected demographic and clinical variables included age, years of education, collection site, race/ethnicity, and cancer and treatment information.

Saliva and/or blood samples were collected. Saliva samples were collected using Oragene kits (DNA Genotek, Kanata, ON, Canada); anticoagulated whole blood was collected with EDTA. Frozen EDTA samples and saliva samples at ambient temperature were shipped to Boston University or subsequently to the Indiana University Genetics Biobank to extract DNA, which was shipped frozen in three batches to the Children’s Hospital of Philadelphia, Center for Applied Genomics, where genome-wide association study (GWAS) assays were performed.

GWAS assays were performed using the Affymetrix Axiom Precision Medicine array (Thermo Fischer Scientific, Waltham, MA, USA) for the first two batches and with the Illumina Global Screening Array v2 (Illumina, San Diego, CA, USA) for the third batch. Microarray data were converted to PLINK format using Illumina GenomeStudio version 2.0 software (Illumina, Inc., San Diego, CA, USA), and processed and quality-controlled with PLINK v1.9 [17]. In total, 807 white non-Hispanic participants had genotype data passing quality control imputed with the haplotype Reference Consortium (HRC) panel using the Michigan Imputation Server [18,19] (see Supplementary Methods in Appendix A for more details). The final data set included 7,661,137 SNPs, >10× the original number of SNPs obtained from genotyping. Of the 807 participants with imputed data, 131 participants were excluded for lack of one-year cognitive data, 4 were excluded due to missing covariates, and 7 were excluded from analysis as they were identified as duplicates or first-degree siblings in the identity-by-descent analysis.

*Apolipoprotein E* (*APOE*) genotype was also obtained separately using TaqMan assays of rs429358 and r7412 on a Real-Time PCR System (Life Technologies, Carlsbad, CA, USA), and/or Fluidigm genotyping with a custom-designed 96-SNP microarray (Fluidigm, San Francisco, CA, USA).

### 2.3. Statistical Methods and Analyses

Genetic data was analyzed in PLINK v1.9 [17]. Primary analyses investigated the interaction of cancer case/control status with genotypes on one-year cognitive performance, controlling for baseline performance. Linear regression was used to predict one-year APE and LM scores based on the main effects of SNPs, group (cancer patient/non-cancer control), and SNP*group interaction, controlling for baseline cognitive scores, age, WRAT4 score, and recruitment site. Sensitivity analysis was performed to investigate the potential influence of *APOE* e4 carrier status; all models were run with/without *APOE* e4 as a covariate. Genomic inflation was calculated for both sets of GWAS results; for APE score analysis, λ = 0.993. For LM analysis, λ = 1.015.

SNP*case/control association analysis results were analyzed with The Functional Mapping and Annotation of Genome-Wide Association Studies (FUMA GWAS) program v1.3.6a [20] (see Supplementary Methods in Appendix A for more information). For SNPs passing the genome-wide significance threshold (*p* < 5 × 10^−8^), the most significant SNP from each locus showing an interaction with cancer group associated with cognitive performance was run in a general linear model in SPSS Statistics 25 (IBM SPSS Statistics 25, IBM Corp., Somers, NY, USA), with interacting term cancer group, dependent factor APE one-year visit, and covariates baseline APE, age, and baseline WRAT4 to calculate marginal means for cancer case/control and carrier groups. Study site was entered as a fixed effect. The model was run with the interaction term SNP*cancer group as well as main effects for all terms, using a Type III sum of squares model including the intercept. Results included marginal means, standard deviations, and upper and lower bounds for the 95% confidence interval, as well as the F statistic and *p*-value for each SNP.

Secondary analyses of all 10,678 genes for enrichment of SNPs within a gene showing interaction with cancer case/control status associated with cognitive performance were also performed in FUMA using the MAGMA program v1.08 [17,20,21,22]. Visualization of results in FUMA included generation of regional SNP plots using data from CADD [23] and RegulomeDB [24]. For gene-level results, significance cut-off was *p* < 5 × 10^−6^.

For lead SNPs at significant loci from the GWAS analyses, we also performed post-hoc testing of the SNPs for quantitative trait loci (QTLs) using the GTEx portal (gtexportal.org/, accessed on 1 March 2023) to investigate the functional consequences of each SNP on gene expression, splicing, and cell-specific regulation of gene expression [25]. For the intergenic SNP identified in the GWAS analysis, we investigated whether this locus was a predicted binding site for any transcription factors using JASPAR [26], a database of transcription factor binding profiles.

## 3. Results

Participants were, on average 68 years old (range 61 to >90), and case and control groups had >15 years of mean education (Table 1). Differences in education and WRAT4 scores between cases and controls were not clinically meaningful. There were no significant differences between women with breast cancer and controls for *APOE* e4 allele frequency. While APE performance was different at baseline and at one-year post-treatment, LM performance was not significantly different in this subset of the total TLC cohort. However, we observed a trend for worse performance in cases than controls and analyzed both scores given that each domain has been significantly associated with CRCD [3].

### 3.1. GWAS and Gene Analyses

#### 3.1.1. GWAS Analyses

Two loci, on chromosomes 1 (rs76859653, *p* = 1.624 × 10^−8^, partial Eta squared = 0.048 for SNP*diagnosis) and 2 (rs78786199, *p* = 1.925 × 10^−8^, partial Eta squared = 0.047 for SNP*diagnosis), were differentially associated with longitudinal APE performance in breast cancer cases compared to controls (see Figure 2A GWAS Manhattan plots for genome-wide analysis results, Appendix A for GWAS QQ plots, Figure 3 and Appendix A for plots of each SNP locus). For rs76859653, there were 329 control noncarriers, 9 control carriers, 319 case noncarriers, and 5 case carriers. For rs78786199, there were 322 control noncarriers, 18 control carriers, 317 case noncarriers, and 7 case carriers. As shown in Figure 4 and Table 2, control individuals carrying minor alleles for either SNP have similar or greater APE one-year mean scores compared to non-carriers controlling for baseline scores. In contrast, cases carrying minor alleles for either of these SNPs have lower APE one-year mean scores than non-carriers controlling for baseline scores, suggesting that in cancer patients but not controls, carriers for either SNP have a greater risk for cognitive decline over time. The analysis of LM domain performance did not identify any SNPs of genome-wide significance (*p* < 5 × 10^−8^, Figure 2B, Appendix A).

For rs76859653 and rs78786199, GTex analysis showed no significant QTLs for either SNP. While this analysis did not identify any genes with differential expression associated with these SNPs, investigation of intergenic rs78786199 with JASPAR showed that this SNP is within the region of predicted transcription factor binding sites for Zinc Finger Imprinted 3, Interferon Regulatory Factors 1, 4, 7, and 8, Signal Transducer and Activator of Transcription 2, and Zinc Finger Protein 317.

Analyses performed with *APOE* e4 carrier status as an additional covariate did not differ significantly.

#### 3.1.2. Gene Level Analyses

Gene analysis did not identify any genes enriched for variants significantly associated (*p* < 5 × 10^−6^) with APE one-year score controlling for baseline when comparing cancer patients to controls (see Figure 5A for gene analysis results, Appendix A for gene QQ plots). However, it is interesting to note that the results for this analysis with the lowest *p*-values were phosphodiesterase 3A (*PDE3A*, *p* = 5.77 × 10^−6^) and phosphodiesterase 4B (*PDE4B*, *p* = 1.49 × 10^−5^), both of which are members of the cyclic AMP-specific cyclic nucleotide phosphodiesterase (PDE) family.

Gene-based analysis of LM domain performance identified one gene, POC5 centriolar protein (*POC5*), which was significantly enriched for variants associated with differences in cognitive performance in cases compared to controls (*p* = 1.99 × 10^−6^, see Figure 5B for gene analysis results, and Appendix A for gene QQ plots).

## 4. Discussion

This first genome-wide study of the association between genetic variation and longitudinal cognitive performance in older women with breast cancer and controls identified two novel loci and three genes of interest. The finding that women with cancer perform differently on cognitive assessments than controls based on minor allele carrier status suggests that genetic background may influence risk for CRCD or may play a role in cognitive dysfunction following diagnosis and treatment.

These analyses identified two loci differentially associated with APE performance over time in cases and controls. The chromosome 1 SNP rs76859653 is intronic (non-coding) in the *HMCN1* gene. This gene encodes an extracellular protein of the immunoglobulin superfamily, suggesting that immune function may play a role in CRCD. The role in humans is unknown, but HMCN in *C. elegans* is involved in maintenance of cell polarity as well as cell migration and invasion [27]. Mutations in this gene have also been identified in gastric and colorectal cancers [28]. Interestingly, a mutation in *HMCN1* has been linked to occurrence of age-related macular degeneration [29,30]. Age-related macular degeneration occurs with an increased risk of dementia [31], providing a potential connection between this gene and CRCD. Computational analysis of potential functions for this SNP including expression quantitative trait analyses did not reveal an obvious mechanism of SNP function; more work will be required to validate this finding and investigate the underlying molecular mechanisms for this locus in CRCD.

The second SNP identified in this analysis, rs78786199, occurs in an intergenic region of chromosome 2. The closest gene to this locus is forkhead box N2 (FOXN2), which is ubiquitously expressed and has been shown to suppress cancer proliferation and invasion [32]. Downregulation of this gene has been shown in acute myeloid leukemia, and was correlated with complex cytogenetic abnormalities [33]. Knockdown of this transcription factor was also associated with increased cancer cell proliferation, an impaired DNA damage response, and chromosomal instability [33]. While FOXN2 has not been specifically identified in neurodegenerative disease or cognitive functional research, perturbations in these molecular pathways have been identified in Alzheimer’s disease as well as cancer [34]. Additionally, rs78786199 is in a predicted binding site for several transcription factors, including Zinc Finger Imprinted 3, Interferon Regulatory Factors 1, 4, 7, and 8, Signal Transducer and Activator of Transcription 2, and Zinc Finger Protein 317. These transcription factors regulate numerous cellular processes, including hematopoiesis, inflammation, immune responses, cell proliferation, regulation of the cell cycle, and induction of growth arrest and programmed cell death in response to DNA damage, suggesting multiple mechanisms that could connect this locus with cancer and CRCD [35,36,37,38,39,40]. The location and current lack of validation for transcription factor binding at this intergenic locus makes functional interpretation challenging, however; more work is required to investigate and validate the molecular mechanisms underlying this locus.

Both SNPs show declining cognitive performance in cancer cases carrying minor alleles, in contrast to stable to improved performance in controls, suggesting that these loci increase risk for CRCD in older women who have experienced breast cancer and its treatment(s). While these findings should be interpreted with caution given the relatively small sample size and low minor allele frequency of the identified genetic variants, the identification of genome-wide significant interactions of SNP or gene and diagnosis with CRCD supports the importance of this research and highlights the need for future studies. Given that these loci were identified in a fairly homogeneous cohort of older white, female, well-educated breast cancer survivors, it seems reasonable to postulate that further study with larger, more diverse cohorts may uncover additional genetic risk factors for CRCD, providing tools for assessment of risk of cognitive decline in cancer survivors and/or avenues for therapeutic research.

Gene-level analysis identified *POC5*; this gene was enriched for variants differentially associated with LM performance change in cases and controls. There is evidence that the POC5 protein is involved in breast cancer cell proliferation and tumorigenesis [41]. POC5 is required for proper assembly of centrioles prior to cell division [42]. A study of histone deacetylases, dysregulation of which can result in carcinogenesis, showed that one mechanism of histone deacetylase action in cancer is to protect POC5 from degradation, resulting in cell cycle progression of cancer cells [41]. Mutations in *POC5* have been identified in adolescent idiopathic scoliosis (AIS); in vitro studies of these mutations have shown that they alter centrosome protein interactions, induce ciliary retraction, and impair cell-cycle progression [43]. A mutation in this gene has also been associated with retinitis pigmentosa [44], an inherited form of retinal degeneration. Inherited retinal disease has extremely heterogeneous genetic etiologies, and syndromic forms can manifest with other symptoms including neurodegeneration, again providing a connection for this gene’s association to cognitive dysfunction as well as cancer [45]. However, much work remains to validate this finding and investigate the molecular mechanisms underlying *POC5* gene variants’ association with CRCD.

The TLC study is unique in having a larger number of individuals with genetic data than prior small studies, though the TLC is still relatively small compared to large scale, well powered sample sizes designed to detect genetic association. Most studies of CRCD are too small to perform any genome-wide analyses and are not powered to use for replication of rarer variants. Therefore, the ability to replicate this study is limited by the lack of well-powered studies of CRCD, particularly in older individuals who may be at increased risk. While there are a number of studies of older individuals with dementia that have cognitive data, these studies typically do not include well-documented cancer history or do not have sufficient populations of breast cancer survivors. These studies typically occur at much later time points following cancer diagnosis than a study of CRCD, making it difficult to meaningfully test for replication. The small number of minor allele carriers for both SNPs reaching genome-wide significance also limited the follow-up that could be done and requires additional replication to increase confidence. Another consideration was the lack of power to specifically examine treatment-related effects; due to the relatively small number of patients treated with chemotherapy (*N* = 84), we were not able to separately assess the impact of cancer and treatment. Larger studies will be required to examine these factors separately. An additional limitation was the exclusion of non-white participants to reduce population-driven genetic bias. This study was not powered to investigate the potential impact of genetic risk factors to CRCD in minority participants. It will be important for future studies to perform similar investigations in other racial and ethnic populations. Specifically, it is critical for additional studies to enroll larger cohorts of non-white participants, to enable the study of race-specific genetic factors underlying CRCD. Once enough samples/studies are available, it will be possible to perform meta-analyses including these data sets to investigate whether these or other genetic risk factors influence the risk for CRCD in minority populations.

## 5. Conclusions

These findings provide preliminary evidence that novel genetic loci may increase susceptibility to CRCD in cancer patients. While these initial results should be interpreted with caution, this study highlights the need for large, well-powered studies of the genetics of CRCD, particularly in older individuals who also have a greater risk for neurodegenerative disease and dementia. As more studies are funded to investigate this critical gap in understanding, it will be possible to perform meta-analyses with existing studies, similar to efforts to increase genetic sample size in Alzheimer’s and Parkinson’s disease research [46,47]. As larger studies become available, we expect meta-analyses to uncover additional genetic factors underlying CRCD, which may be used to further inform patient management and therapeutic research.

## Figures and Tables

**Figure 1 cancers-15-02877-f001:**
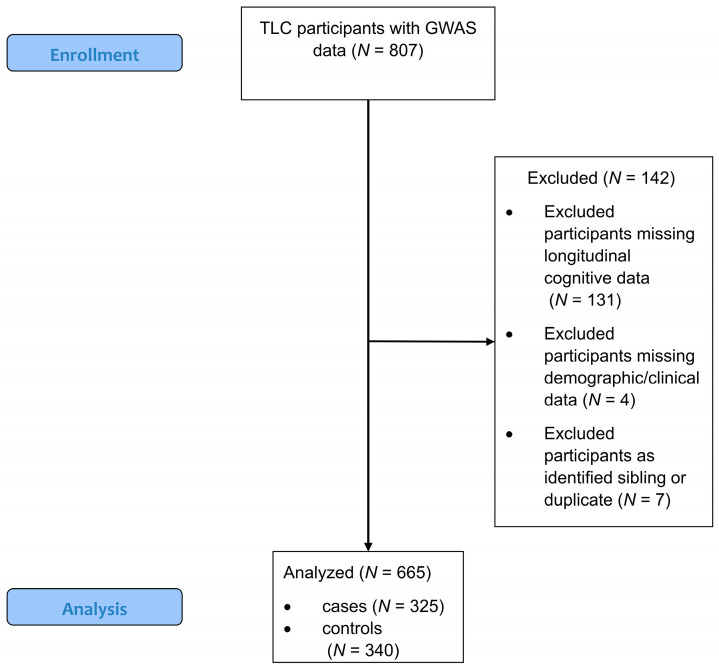
CONSORT Diagram. A total of 807 White non-Hispanic participants had imputed GWAS data passing quality control. Of these, 142 were removed due to missing longitudinal data (131 for missing cognitive performance data, 4 for missing demographic/clinical information, and 7 for duplicate or first-degree sibling status). The final data set for analysis included 665 individuals, 325 cases and 340 controls.

**Figure 2 cancers-15-02877-f002:**
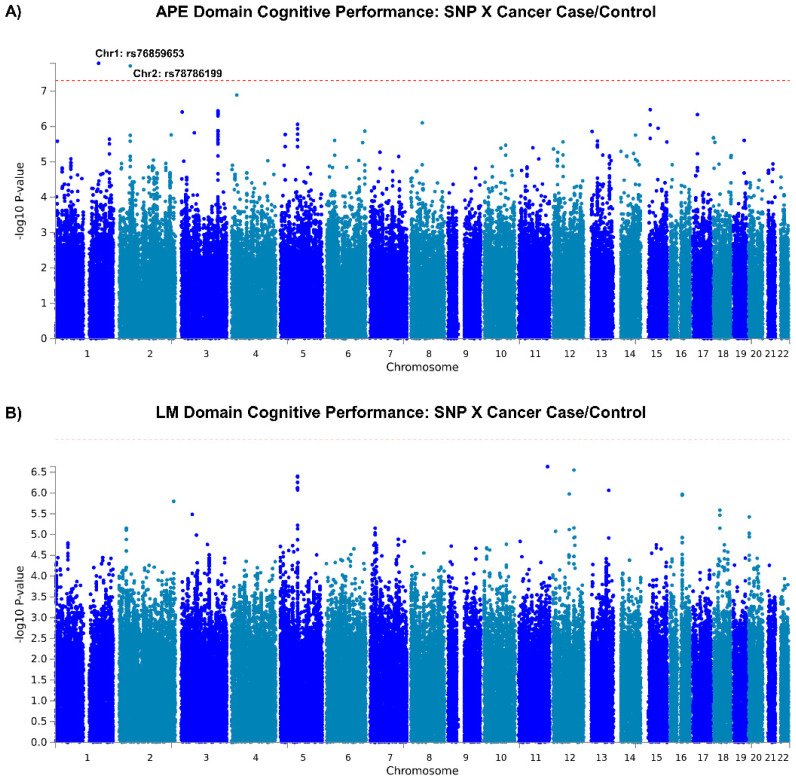
GWAS SNP*Cancer Interaction. (**A**) Manhattan plot of GWAS SNP by group interaction associated with one-year follow-up (1Y) visit attention, processing speed, and executive function (APE) cognitive domain score. GWAS genome-wide analysis of SNP*group (0/1) interaction with outcome of 1Y visit APE score, covarying for age, baseline WRAT4 score, site, and baseline APE score. Loci on chromosomes 1 (rs76859653, *p* = 1.624 × 10^−8^) and 2 (rs78786199, *p* = 1.925 × 10^−8^) have *p*-values of genome-wide significance (*p* < 5 × 10^−8^). (**B**) Manhattan plot of GWAS SNP by group interaction associated with 1Y visit learning and memory (LM) cognitive domain score. GWAS genome-wide analysis of SNP*group (0/1) interaction with outcome of 1Y visit LM score, covarying for age, baseline WRAT4 score, site, and baseline LM score. No loci attained genome-wide significance (*p* < 5 × 10^−8^).

**Figure 3 cancers-15-02877-f003:**
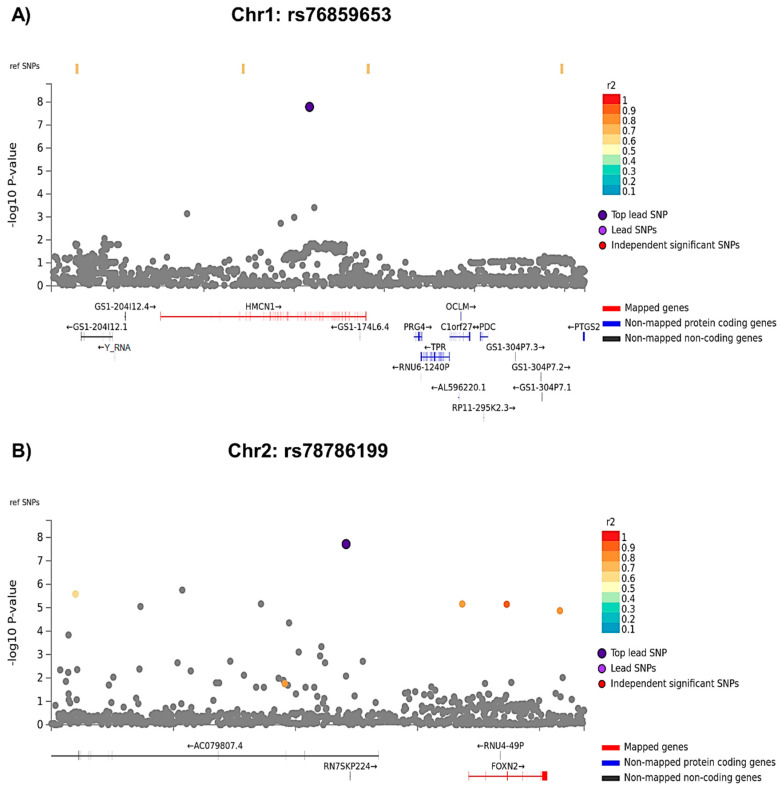
Regional Plots for Lead SNPs from APE GWAS Analysis. Each panel shows the SNPs adjacent to the lead SNP. Top SNP is colored navy; independent significant SNPs are color coded by r2 scale. SNPs that are not in linkage disequilibrium (LD) with the top lead SNP are shown in gray. ‘Ref SNPs’, displayed at the top of the plot, are SNPs that are in LD with the top SNP, but which do not have a *p* value because they were not included in the data. Mapped genes are shown in red. Y-axis shows the −log10 *p*-value of all graphed SNPs. Location on the chromosome in base pairs is shown on the X-axis. (**A**) Regional SNP Plot for rs76859653 on chromosome 1, a lead SNP significantly differently associated with APE cognitive domain performance at one-year follow-up (1Y) visit by group, covarying for age, baseline WRAT4 score, site, and baseline APE performance. (**B**) Regional SNP Plot for rs78786199 on chromosome 2, a lead SNP significantly differentially associated with APE cognitive domain performance at 1Y visit by group, covarying for age, baseline WRAT4 score, site, and baseline APE performance.

**Figure 4 cancers-15-02877-f004:**
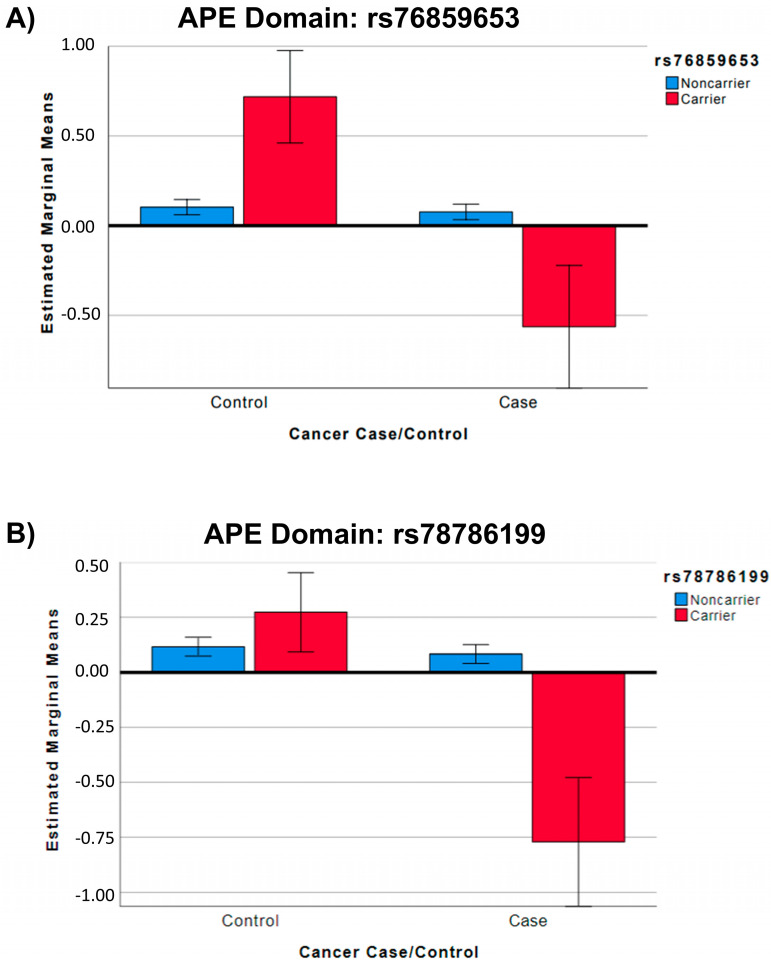
SNPs Differentially Associated with APE Score by Group. Boxplots depicting differences in APE score at one-year follow-up (1Y) visit (Y-axis) by cancer case/control status (X-axis) and by SNP minor allele carrier status (red = carrier, blue = noncarrier). There were no individuals homozygous for either SNP; all carriers are heterozygous. Error bars indicate 95% confidence intervals. Covariates appearing in the model were evaluated at the following values: age = 68.05, WRAT4 = 112.33, baseline APE = 0.03, site 1 = 1.21, site 2 = 1.17, site 3 = 1.28, site 4 = 1.14, site 5 = 1.07. (**A**) Results for rs7659653. There were 329 control noncarriers, 9 control carriers, 319 case noncarriers, and 5 case carriers. (**B**) Results for rs78786199. There were 322 control noncarriers, 18 control carriers, 317 case noncarriers, and 7 case carriers.

**Figure 5 cancers-15-02877-f005:**
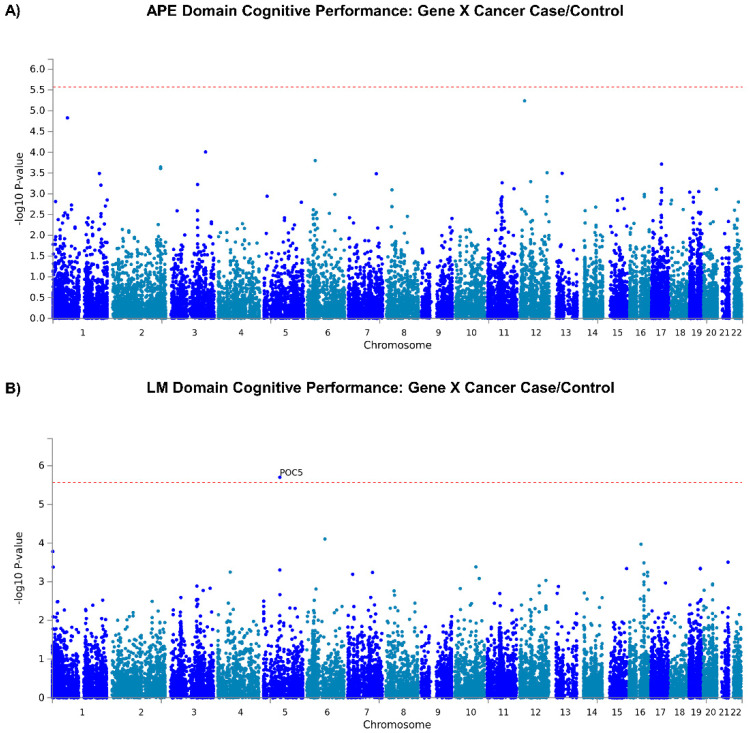
GWAS Gene*Group Interaction. SNPs shown by chromosomal location (X-axis); chromosomes displayed in alternating blue/gray colors (i.e., SNPs on chromosome 1 are blue, SNPs on chromosome 2 are gray, etc.). (**A**) Manhattan plot of TLC gene by group interaction associated with one-year follow-up (1Y) visit attention, processing speed, and executive function (APE) cognitive domain score. GWAS genome-wide analysis of gene*group (0/1) interaction with outcome of 1Y visit APE score, covarying for age, baseline WRAT4 score, site, and baseline APE score. No genes attained genome-wide significance (*p* < 5 × 10^−6^). (**B**) Manhattan plot of TLC GWAS gene by group interaction associated with 1Y visit learning and memory (LM) cognitive domain score. GWAS genome-wide analysis of gene*group (0/1) interaction with outcome of 1Y LM score, covarying for age, baseline WRAT4 score, site, and baseline LM score. One gene, POC5 centriolar protein (*POC5*), attained genome-wide significance (*p* = 1.99 × 10^−6^).

**Table 1 cancers-15-02877-t001:** Demographics.

Variable	Case (*N* = 325) *	Control (*N* = 340)	*p*-Value **
Age, mean years (StDev)	68.2 (5.7)	67.9 (6.6)	*0.596*
Education, mean years (StDev)	15.3 (2.1)	15.7 (2.2)	* **0.012** *
WRAT4 score, mean (StDev)	111.0 (15.8)	113.7 (15.4)	* **0.028** *
Chemotherapy treatment, number (%)	84 (25.8%)	-	*-*
Hormone therapy, number (%)	257 (79.1%)	-	*-*
APOE e4 carrier, number (%)	81 (24.9%)	85 (25.0%)	*1.000*
APE baseline score, mean (StDev)	−0.034 (0.634)	0.094 (0.608)	* **0.008** *
APE one-year score, mean (StDev)	0.016 (0.643)	0.173 (0.607)	* **0.001** *
LM baseline score, mean (StDev)	0.013 (0.792)	0.056 (0.811)	*0.487*
LM one-year score, mean (StDev)	0.153 (0.839)	0.238 (0.607)	*0.185*

StDev = standard deviation; WRAT4 = Wide Range Achievement Test-Fourth Edition Word Reading Test; *APOE* = apolipoprotein E; APE = Attention, Processing speed, and Executive function; LM = Learning and Memory. * 84 (25.8%) of the 325 cases were treated with chemotherapy between the baseline and one-year visits. ** *p*-values are italicized, with values <0.05 on ANOVA or Fischer’s Exact two-sided test shown in bold.

**Table 2 cancers-15-02877-t002:** SNPs associated with longitudinal attention, processing speed, and executive function (APE) domain scores differentially in cancer cases vs. controls.

SNP	Group	SNP MA *	APE Mean ** (StE)	95% CI Lower Bound	95% CI Upper Bound	F	*p* Value **
rs76859653	Control	0 (*N* = 329)	0.10 (0.02)	0.06	0.15	32.68	<0.001
	Control	1 (*N* = 9)	0.71 (0.13)	0.46	0.97		
	Case	0 (*N* = 319)	0.08 (0.02)	0.03	0.12		
	Case	1 (*N* = 5)	−0.56 (0.17)	−0.91	−0.22		
rs78786199	Control	0 (*N* = 322)	0.12 (0.02)	0.07	0.16	32.27	<0.001
	Control	1 (*N* = 18)	0.27 (0.09)	0.09	0.45		
	Case	0 (*N* = 317)	0.08 (0.02)	0.04	0.13		
	Case	1 (*N* = 7)	−0.77 (0.15)	−1.06	−0.48		

StE = Standard Error; CI = confidence interval. * 0 = homozygous for major allele, 1 = heterozygous for minor allele; neither of the top SNPs have any homozygous minor allele genotypes in the data set. Note, results are listed for all participants with genotypes passing QC for the SNPs of interest. ** Results from general linear models with individual significant SNPs identified from GWAS analysis (APE one-year score ~ SNP*group + baseline APE score + age + WRAT4 score + recruitment site); cognitive means for APE score calculated controlling for mean-centered covariates.

## Data Availability

All data is available on request from the corresponding author. A pre-print version of this manuscript is available on MedRxiv (https://www.medrxiv.org/content/10.1101/2022.09.12.22279861v1.full).

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
