# Peer review of "Genetic Variants Associated with Longitudinal Cognitive Performance in Older Breast Cancer Patients and Controls†"

_cancers, 2023, doi:10.3390/cancers15112877_

Round 1

Reviewer 1 Report

Thank you for this contribution, generating new knowledge to our understanding of some mechanisms for the development of CRCI. Your study is the first genome-wide assessment of CRCI in women with breast cancer. While the authors note some limitations to their findings, in general the sample size is adequate for the analyses and interpretation thereof.  These findings will support the potential for identifying those at risk for development of CRCI at time of diagnosis and may prompt early intervention.

This is an extremely well-written manuscript with appropriate references and supplementary materials.  The abstract is concise and clearly conveys the importance of their findings. The introduction and supportive literature for performing the study reflects how the findings may impact our understanding for the development of CRCI.  The analyses are appropriate and well written.  Finally, the results are comprehensive and appropriate interpretation that it outlined in the discussion.

I do not have any mods to suggest.

Author Response

Thank you very much for the positive feedback; we are excited to share these results with the research community! 

Reviewer 2 Report

The authors conducted a GWAS study, with prospective longitudinal data collected from approximately 300 breast cancer patients and approximately 300 age and education matched controls, to identify genetic markers associated with cancer-related cognitive decline (CRCD) prior to systematic treatment and at 1 year follow up.

The study is interesting and well presented. This reviewer identified few minor points for improvement:

1) Information about the measurement of the phenotypes (APE and LM)  should be presented in the text, not only in the Supplementary materials.

2) Differences in education and WRAT4 score were statistically significant between cases and controls (Table 1). Can the authors please clarify how did they determine that they were not clinically significant?

3) Figure 3 was too small - this reviewer could not identify any information based on the text and the footnote.

4) Lines 265-267 present information on how many cases and how many controls were heterogyzous for the minor allele. This reviewer suggests that this information is presented in the text.

Author Response

We would like to thank the reviewer for these comments, as they have improved the clarity of the results.  We have included a point by point response below:

1) Information about the measurement of the phenotypes (APE and LM)  should be presented in the text, not only in the Supplementary materials.

This information has been moved from the supplement to the text (lines 127-133).

2) Differences in education and WRAT4 score were statistically significant between cases and controls (Table 1). Can the authors please clarify how did they determine that they were not clinically significant?

The difference of 2.7 points between groups in WRAT4 word reading score is less than 1/5 of a standard deviation. Mean scores for both groups are in the lower end of the high average range, and at comparable grade level equivalents (post-high school). The difference in education is 0.4 year (less than a semester difference; 15.3 vs. 15.7 years), indicating that both groups have a mean level of education just under a 4-year college degree. Three expert neuropsychologists on our team (AJS, BCM, JCR) reviewed these data, and concur that these differences are not clinically meaningful.

3) Figure 3 was too small - this reviewer could not identify any information based on the text and the footnote.

To address this issue, the figure has been split – the zoom plot showing SNPs in LD at each locus is still in the main text, but the size and resolution have been increased.  The CADD and RegulomeDB graphs for each locus have been moved to the supplement (Figure 2S) and the size and resolution have been increased.

4) Lines 265-267 present information on how many cases and how many controls were heterozygous for the minor allele. This reviewer suggests that this information is presented in the text.

This information has been added to the text on lines 220-222.